# Development of an Automated Visibility Analysis Framework for Pavement Markings Based on the Deep Learning Approach

**Kyubyung Kang** [1] , **Donghui Chen** [2] , **Cheng Peng** [2] , **Dan Koo** [1] , **Taewook Kang** [3,*] and **Jonghoon Kim** [4]

[1] Department of Engineering Technology, Indiana University-Purdue University Indianapolis (IUPUI), Indianapolis, IN 46038, USA; kyukang@iu.edu (K.K.); dankoo@iu.edu (D.K.)
[2] Department of Computer and Information Science, Indiana University-Purdue University Indianapolis (IUPUI), Indianapolis, IN 46038, USA; dch1@iu.edu (D.C.); cp16@iu.edu (C.P.)
[3] Korea Institute of Civil Engineering and Building Technology, Goyang-si 10223, Korea
[4] Department of Construction Management, University of North Florida, Jacksonville, FL 32224, USA; jongkim@unf.edu
[*] Correspondence: ktw@kict.re.kr; Tel.: +82-10-3008-5143

**Abstract:** Pavement markings play a critical role in reducing crashes and improving safety on public roads. As road pavements age, maintenance work for safety purposes becomes critical. However, inspecting all pavement markings at the right time is very challenging due to the lack of available human resources. This study was conducted to develop an automated condition analysis framework for pavement markings using machine learning technology. The proposed framework consists of three modules: a data processing module, a pavement marking detection module, and a visibility analysis module. The framework was validated through a case study of pavement markings training data sets in the U.S. It was found that the detection model of the framework was very precise, which means most of the identified pavement markings were correctly classified. In addition, in the proposed framework, visibility was confirmed as an important factor of driver safety and maintenance, and visibility standards for pavement markings were defined.

**Keywords:** pavement markings; deep learning; visibility; framework

## 1. Introduction

Pavement markings play a critical role in reducing crashes and improving safety on public roads. They do not only convey traffic regulations, road guidance, and warnings for drivers, but also supplement other traffic control devices such as signs and signals. Without good visibility conditions of pavement markings, the safety of drivers is not assured. Therefore, it is important for transportation agencies and other stakeholders to establish a systematic way of frequently inspecting the quality of pavement markings before accidents occur.

State highway agencies in the U.S. invest tremendous resources to inspect, evaluate, and repair pavement markings on nearly nine million lane-miles [1]. One of the challenges in pavement marking inspection and maintenance is the variable durability of pavement markings. Conditions of pavement markings vary even if they were installed at the same time. Such conditions are highly dependent on the material characteristics, pavement characteristics, traffic volumes, weather conditions, etc. Unfortunately, inspecting all pavement markings at the right time is very challenging due to the lack of available human resources. Hence, an automated system for analyzing the condition of pavement markings is critically needed. This paper discusses a study that developed an automated condition

analysis framework for pavement markings using machine learning technology. The proposed framework consists of three modules: a data processing module, a pavement marking detection module, and a visibility analysis module. The data processing module includes data acquisition and data annotation, which provides a clear and accurate dataset for the detection module to train. In the pavement marking detection module, a framework named YOLOv3 is used for training to detect and localize pavement markings. For the visibility analysis module, the contour of each pavement marking is clearly marked, and each contrast intensity value is also provided to measure visibility. The framework was validated through a case study of pavement markings training data sets in the U.S.

## 2. Related Studies

With the remarkable improvements in cameras and computers, pavement conditions can now be analyzed remotely using image processing technologies. Unlike traditional manual inspection, remote analysis does not require on-site operations and closed traffic, yet has high inspection accuracy and efficiency, which greatly reduces the management costs of the government's transportation department. With the continuous development of computer vision technology, more and more researchers are exploring how to use videos or images to complete the analysis of pavement systems. Ceylan et al. summarized the recent computer vision-based pavement engineering applications into seven categories: estimation of pavement conditions and performance, pavement management and maintenance, pavement distress prediction, structural evaluation of pavement systems, pavement image analysis and classification, pavement materials modeling, and other transportation infrastructure applications [2,3]. The increasing number of publications and technologies in these fields in recent years undoubtedly demonstrates that more and more researchers are interested in exploring the use of computer vision technology to study pavement engineering problems [4–10].

A complete pavement system consists mainly of the pavement and the painted markings or lanes. Intuitively, most studies of pavement systems focused on analyzing the pavements and markings. Regarding pavements, researchers pay more attention to exploring how to efficiently and precisely detect cracks on roads. Traditional techniques start mainly from pattern matching or texture analysis to help locate cracks. However, due to the diversity of cracks and their unfixed shapes, such traditional techniques have been found wanting. Studies have been conducted on the automatic identification of pavement cracks using neural network algorithms, due to their powerful learning and representing capabilities. In this new technique, the characteristic information on the road images is first extracted, and then the neural network is trained to recognize it. For sample pavements with known characteristics, the neural network can automatically learn and memorize them, whereas for unknown pavement samples, the neural network can automatically make inferences based on previously learned information.

Meignen et al. directly flattened all the pixels of each image into one-dimensional feature vectors, which were taken as the inputs to a neural network [11]. This method did not work very well, as different roads had different crack characteristics, and the training input data set was too large. Therefore, it is wise to first extract the features that are meaningful for recognizing pavement cracks, and then process the features using a neural network. Xu et al. proposed a modified neural network structure to improve the recognition accuracy [8]. First, the collected pavement images were segmented into several parts, after which the features were extracted from each part. For each segment, the probability that it could have cracks was inferred with the neural network model. The regional division strategy reduced the differences between the samples and effectively improved the performance of the network. Zhang et al. trained a supervised convolutional neural network (CNN) to decide if a patch represents a crack on a pavement [10]. The authors used 500 road system images taken with a low-cost smartphone to inspect the performance of the proposed model. The experiment results showed that the automatically learned features of the deep CNN provided a superior crack recognition capability compared with the features extracted from the hand-craft approaches. Li et al. explored the possibility that the size of the reception field in the CNN structure influences its performance [6]. They trained four CNNs with

different reception field sizes and compared them. The results showed that the smaller reception fields had slightly better model accuracy, but also had a more time-consuming training process. Therefore, a good trade-off between effectiveness and efficiency was needed. In the study of Zhang et al., a recurrent neural network (RNN) named CrackNet-R was modeled to perform automatic pixel-wise crack detection for three-dimensional asphalt pavements [12]. In this model, the authors applied a new structure, a gated recurrent multilayer perceptron network, which showed a better memorization ability than other recurrent schemes. Relying on such a memorization ability, the CrackNet-R first searched the image sequence with the highest probability of having a crack pattern. Then an output layer was adopted to transform the timely probabilities of the sequence into pixel-wise probabilities. This novel pixel-wise pavement crack detection model provided a new orientation for the development of the field.

For pavement markings, many publications have also focused on the detection and classification of road signs or lanes, which is an important task for pavement system maintenance or autonomous driving. Most previous studies on this problem were developed with the image processing theory and the hand-craft pattern functions, which made it very difficult to generalize in various situations. Chen et al. proposed a two-stage framework for road markings detection and classification based on machine learning [13]. The first-stage detection model was carried out with the binarized normed gradient (BING) approach, and the second-stage classification model was realized with the principal component analysis network (PCANet). Both BING and PCANet are popular techniques in the field of machine learning. Yamamoto et al. adopted a simple neural network to recognize pavement markings on road surfaces [9]. The authors first extracted the candidate road areas based on the edge information, and then fed them to the neural network to accomplish the recognition. Gurghian et al. proposed a novel method called DeepLanes to directly estimate, from images taken with a side-mounted camera, the position of the lanes using a deep neural network [14]. Besides the ability of the proposed model to determine the existence of lane markings, it could also predict the position of the lane markings with an average speed of 100 frames per second at the centimeter level, without any auxiliary processing. This algorithm can provide significant support for the driver-assistant system that depends on the lanes. The aforementioned models mainly treated pavement markings and lanes as different objects for processing and analysis, until the emergence of the vanishing point guided network (VPGNet), which Lee et al. proposed [15]. VPGNet was an end-to-end deep CNN inspired by the multi-task network structure that can simultaneously detect road markings and lanes. It introduced the vanishing point prediction task into the network to guide lane detection, which improved the performance of the network in some bad situations such as rainy days or at night. The authors also provided a public image dataset for lane and road marking detection tasks with pixel-wise annotations.

## 3. Methodology

Figure 1 shows an overview of the proposed framework with its three modules: a data processing module, a pavement marking detection module, and a visibility analysis module.

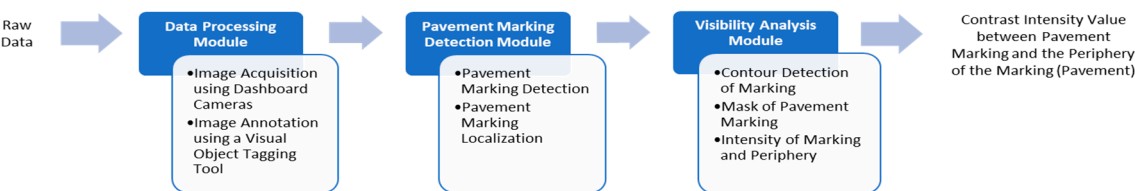

**Figure 1.** Overview of the Framework for Condition Analysis of Pavement Markings.

### 3.1. Data Processing Module

#### 3.1.1. Data Acquisition

Since deep learning is a kind of data-driven algorithm, a training dataset must be prepared for the network model. Due to the quick development of autonomous driving, many public datasets for each driving situation have been collected, such as BDD100K, KITTI, Caltech Lanes, and VPGNet [15–18]. However, these datasets mainly focus on lane detection rather than the pavement markers. The VPGNet dataset provides annotations for lanes and pavement markers, but its pixel-wise annotations are inappropriate for the detection module used in this study. Thus, a system for automatically gathering images or videos of pavement systems must be set up. An action camera mounted behind the front windshield of a car driving on the roadways of Indianapolis, U.S.A. was used to record high-definition (HD) videos. Generally, the camera can capture 90% of the view in front of the moving vehicle, including pavements, transportation systems, and nearby street views, but only the data on pavements were used in this study. For this study, several trips were taken to record plenty of video data. The collected dataset covered various weather conditions, such as daytime, nighttime, sunny days, and rainy days, and different regions such as highways and urban areas. After screening all the video data, more than 1000 high-quality pictures were intercepted, of which about 200 were used for testing, and the remaining pictures were used for training, which maintained a good training-testing ratio.

#### 3.1.2. Data Annotation

Since the primary goal of this study was to make the computer recognize the pavement markings in the road view videos or images, a labeled dataset had to be prepared for the model training process. After comparing multiple open-source labeling software, the visual object tagging tool (VoTT) was chosen to perform the data annotation. VoTT is a powerful open-source labeling software released by Microsoft [19]. This software provides a technique for automatic labeling based on the pre-trained network, which can significantly reduce the workload for annotations. It also supports many formats of the exported annotation results, which make the labeled sample set suitable for various deep learning development frameworks. Figure 2 shows an example of the labeling process.

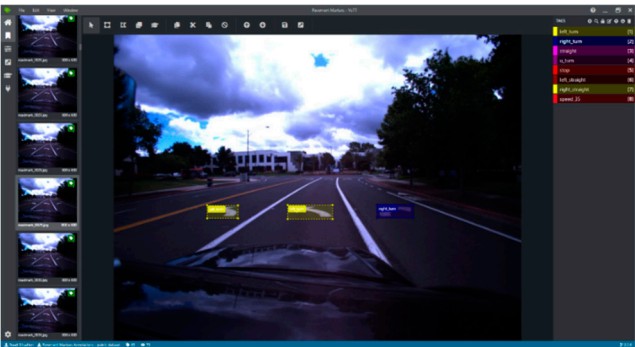

**Figure 2.** The software interface for data annotation tasks.

The VOC XML file format was chosen to generate annotations for each imported image. The key step in this procedure is to develop categories for the pavement markings. This study mainly focused on arrow-like pavement markings, such as those for left turn, right turn, etc. However, up to 10 categories of pavement markings were additionally captured with rectangular boxes for future research. Those categories are described in Figure 3. The annotated data were divided into a training dataset and a testing dataset at a ratio of 0.9:0.1.

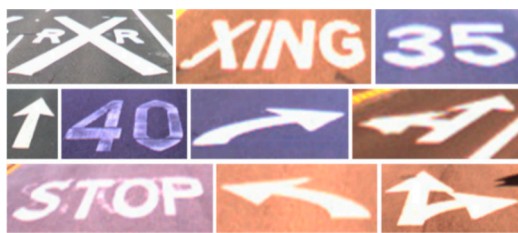

**Figure 3.** Types of labeled pavement markings.

### 3.2. Pavement Markings Detection Module

In the field of computer vision, many novel object recognition frameworks have been studied in recent years. Among these frameworks, the most studied frameworks are deep learning-based models. Generally, according to the recognition principle, existing object detection models can be divided into two categories: two-stage frameworks and one-stage frameworks [20].

In two-stage frameworks, the visual target is detected in mainly two steps. First, abundant candidate regions that can possibly cover the targets are proposed, and then the validity of such regions is determined. R-CNN, Fast-RCNN, and Faster-RCNN are the representative two-stage frameworks, all of which have high a detection precision [21–23]. However, since they take time to generate candidate regions, their detection efficiency is relatively unpromising, which makes them unsuitable for real-time applications. To make up for this deficiency, researchers proposed the one-stage framework.

Compared to the two-stage framework, the one-stage framework gets rid of the phase for proposing candidate regions, and simultaneously performs localization and classification by treating the object detection task as a regression problem. Moreover, with the help of CNN, the one-stage framework can be constructed as an end-to-end network so that inferences can be made with simple matrix computations. Although this type of framework is slightly inferior to the two-stage framework in detection accuracy, its detection speed is dozens of times better. One of the representative one-stage frameworks, You Only Look Once (YOLO), achieves a balance between detection accuracy and speed [24]. After continuous updating and improvement, the detection accuracy of YOLOv3 has already caught up with that of most two-stage frameworks. This is why YOLOv3 was chosen as the pavement markings detection model in this study.

#### 3.2.1. Demonstration of the YOLO Framework

Previous studies, such as on Region-CNN (R-CNN) and its derivative methods, used multiple steps to complete the detection, and each independent stage had to be trained separately, which slowed down the execution and made optimizing the training process difficult. YOLO uses an end-to-end design idea to transform the object detection task into a single regression problem, and directly obtains the coordinates and classification probabilities of the targets from raw image data. Although Faster-RCNN also directly takes the entire image as an input, it still uses the idea of the proposal-and-classifier of the R-CNN model. The YOLO algorithm brings a new solution to the object detection problem. It only scans the sample image once and uses the deep CNNs to perform both the classification and the localization. The detection speed of YOLO can reach 45 frames per second, which basically meets the requirement of real-time video detection applications.

YOLO divides the input image into $S * S$ sub-cells, each of which can detect objects individually. If the center point of an object falls in a certain sub-cell, the possibility of including the object in that sub-cell is higher than the possibility of including it in the adjacent sub-cells. In other words, this sub-cell should be responsible for the object. Each sub-cell needs to predict $B$ bounding boxes and the confidence score that corresponds to each bounding box. In detail, the final prediction is a five-dimensional array, namely, $(x, y, w, h, c)^T$, where $(x, y)$ is the offset that compares the center point of the bounding box with the upper left corner of the current sub-cell; $(w, h)$ is the aspect ratio of the bounding box relative to the entire image; and $c$ is the confidence value. In the YOLO framework,

the confidence score has two parts: the possibility that there is an object in the current cell, and the Intersection over Union (IoU) value between the predicted box and the reference one. Suppose the possibility of the existence of the object is $\Pr(Obj)$, and the IoU value between the predicted box and the reference box is $IoU(pred, truth)$, the formula for the confidence score is shown as Equation (1).

$$Confidence = \Pr(Obj) * IoU(pred, truth) \tag{1}$$

Suppose that $box_p$ is the predicted bounding box, and $box_t$ is the reference bounding box. Then the IoU value can be calculated using the following formula.

$$IoU_p^t = \frac{box_p \cap box_t}{box_p \cup box_t} \tag{2}$$

In addition, YOLO outputs the individual conditional probability of $C$ object categories for each cell. The final output of the YOLO network is a vector with $S * S * (5 * B + C)$ nodes.

YOLO adopted the classic network structure of CNN, which first extracted spatial features through convolutional layers, and then computed predictions by fully connected layers. This type of architecture limits the number of predictable target categories, which makes the YOLO model insufficient for multi-object detection. Moreover, since YOLO randomly selects the initial prediction boxes for each cell, it cannot accurately locate and capture the objects. To overcome the difficulties of YOLO and enhance its performance, Redmon et al. further modified its structure, applied novel features, and proposed improved models such as YOLOv2 and YOLOv3 [25,26].

The YOLOv2 network discarded the fully connected layers of YOLO, transformed YOLO into a fully convolutional network, and used the anchor boxes to assist in the prediction of the final detection bounding boxes. It predefined a set of anchor boxes with different sizes and aspect ratios in each cell to cover different positions and multiple scales of the entire image. These anchor boxes were used as initial candidate regions, which were distinguished according to the presence or absence of the targets inside them through the network. The position of the predicted bounding boxes was also continuously fine-tuned [27]. To fit the characteristics of the training samples, YOLOv2 used the k-means clustering algorithm to automatically learn the best initial anchor boxes from the training dataset. Moreover, YOLOv2 applied the Batch Normalization (B.N.) operation to the network structure. B.N. decreased the shift in the unit value in the hidden layer, and thus improved the stability of the neural network [28]. The B.N. regularization can prevent overfitting of the model, which makes the YOLOv2 network easier to converge.

Compared to YOLOv2, YOLOv3 mainly integrated some advanced techniques. While maintaining the fast detection, it further improved the detection accuracy and the ability to recognize small targets. YOLOv3 adopted a novel framework called Darknet-53 as its main network. Darknet-53 contained a total of 53 convolutional layers and adopted the skip-connection structure inspired by ResNet [29]. The much deeper CNN helped improve feature extraction. Motivated by the idea of multilayer feature fusion, YOLOv3 used the up-sampling method to re-extract information from the previous feature maps, and performed feature fusion with different-scale feature maps. In this way, more fine-grained information can be obtained, which improved the accuracy of the detection of small objects.

### 3.2.2. Structure of YOLOv3

Figure 4 shows the YOLOv3 network structure, which has two parts: Darknet-53 and the multi-scale prediction module. Darknet-53 is performed to extract features from the input image, the size of which is set at $416 \times 416$. It consists of two $1 * 1$ and $3 * 3$ convolutional layers, without any fully connected layers. Each convolutional layer is followed by a B.N. layer and a LeakyReLU activation function, which is regarded as the DBL block. In addition, Darknet-53 applies residual blocks in some layers. The main distinction of the residual block is that it adds a direct connection from the block entrance to the block exit, which helps the model to converge more easily, even if the network

is very deep. When the feature extraction step is completed, feature maps are used for multi-scale object detection. In this part, YOLOv3 extracts three feature maps of different scales in the middle, middle-bottom, and bottom layers. In these layers, the concatenation operations are used to fuse the multi-scale features. In the end, three predictions of different scales will be obtained, each of which will contain the information on the three anchor boxes. Each anchor box is represented as a vector of $(5 + num_{class})$ dimensions, in which the former five values indicate the coordinates and the confidence score, and $num_{class}$ refers to the category number of the objects. In this study, five kinds of arrow-like pavement markings were considered.

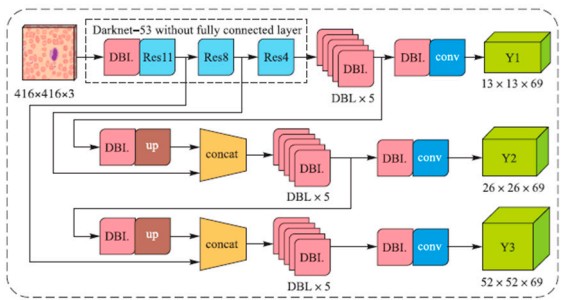

**Figure 4.** Structure of the YOLOv3 network. (https://plos.figshare.com/articles/YOLOv3_architecture_/8322632/1).

### 3.3. Visibility Analysis Module

After the data collected by the dashboard camera are labeled, a YOLOv3-based pavement marking detection module can be constructed and trained. The target pavement markings can be extracted and exported as small image patches. Thus, the next step is to design a visibility analysis module to help determine the condition of the pavement markings.

Pavement markings are painted mainly to give notifications to drivers in advance. As such, a significant property of pavement markings is their brightness. However, brightness is an absolute value affected by many factors, such as the weather and the illumination. Since the human visual system is more sensitive to contrast than to absolute luminance, the intensity contrast is chosen as the metric for the visibility of pavement markings [30]. In this study, contrast was defined as the difference between the average intensity of the pavement marking and the average intensity of the surrounding pavement. The main pipeline of this visibility analysis module is shown in Figure 5.

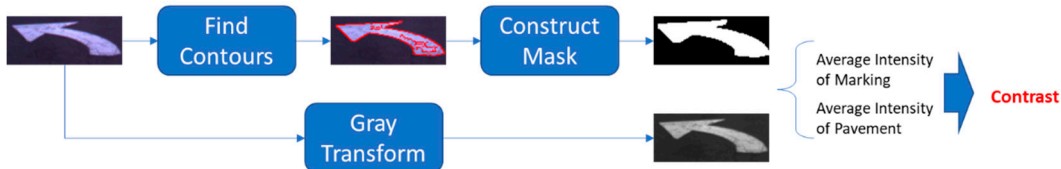

**Figure 5.** A demonstration of the pipeline of the visibility analysis module.

### 3.3.1. Finding Contours

As the pavement markings are already exported as image patches, the first step is to separate the pavement markings from the pavement. Since only arrow-like markings were considered in this study, the portion with the marking can be detached easily from the image, for as long as the outer contour of the marking is found. The contour can be described as a curve that joins all the continuous points along the boundary with the same color or intensity.

The contour tracing algorithm used in this part was proposed by Suzuki et al. [30] It was one of the first algorithms to define the hierarchical relationships of the borders and to differentiate the outer bounders from the hole bounders. This method has been integrated into the OpenCV Library [31].

The input image should be a binary image, which means the image has only two values: 0 and 1, with 0 representing the black background, and 1, the bright foreground or object. Thus, the border should mainly serve as the edge.

Assume that $p_{ij}$ denotes the pixel value at position $(i, j)$ in the image. Two variables, Newest Border Number (*NBD*), Last Newest Border Number (*LNBD*), are created to record the relationship between the pixels during the scanning process. The algorithm uses the row-by-row and left-to-right scanning schemes to process each *NBD* and *LNBD*, where $p_{ij} > 0$.

Step 1. If $p_{ij} = 1$ and $p_{i,j-1} = 0$, which indicate that this point is the starting point of an outer border, increment *NBD* by 1 and set $(i_2, j_2) \leftarrow (i, j - 1)$. If $p_{ij} \geq 1$ and $p_{i,j+1} = 0$, which means it leads a hole border, increment *NBD* by 1 and set $(i_2, j_2) \leftarrow (i, j + 1)$ and *LNBD* $\leftarrow p_{ij}$ in case $p_{ij} > 1$. Otherwise, jump to Step 3.

Step 2. From this starting point $(i, j)$, perform the following operations to trace the border.

2.1. Starting from pixel $(i_2, j_2)$, traverse the neighborhoods of pixel $(i, j)$ in a clockwise direction. In this study, the 4-connected case is selected to determine the neighborhoods, which means only the points connected horizontally and vertically are regarded as the neighborhoods. If a non-zero value exists, denote such pixel as $(i_1, j_1)$. Otherwise, let $p_{ij} = -NBD$ and jump to Step 3.

2.2. Assign $(i_2, j_2) \leftarrow (i_1, j_1)$ and $(i_3, j_3) \leftarrow (i, j)$.

2.3. Taking pixel $(i_3, j_3)$ as the center, traverse the neighborhoods in a counterclockwise direction from the next element $(i_2, j_2)$ to find the first non-zero pixel, and assign it as $(i_4, j_4)$.

2.4. Update the value $p_{i_3,j_3}$ according to Step 2.4 in Figure 6.

2.5. If $p_{i_3,j_3+1} = 0$, update $p_{i_3,j_3} \leftarrow -NBD$.

2.6. If $p_{i_3,j_3+1} \neq 0$ and $p_{i_3,j_3} = 1$, update $p_{i_3,j_3} \leftarrow NBD$.

2.7. If the current condition satisfies $(i_4, j_4) = (i, j)$ and $(i_3, j_3) = (i_1, j_1)$, which means it goes back to the starting point, jump to Step 3. Otherwise, assign $(i_2, j_2) \leftarrow (i_3, j_3)$ and $(i_3, j_3) \leftarrow (i_4, j_4)$ and return to Sub-step 2.3.

Step 3. If $p_{ij} \neq 1$, update *LNBD* $\leftarrow |p_{ij}|$. Let $(i, j) \leftarrow (i, j + 1)$ and return to Step 1 to process the next pixel. This algorithm stops after the most bottom-right pixel of the input image is processed.

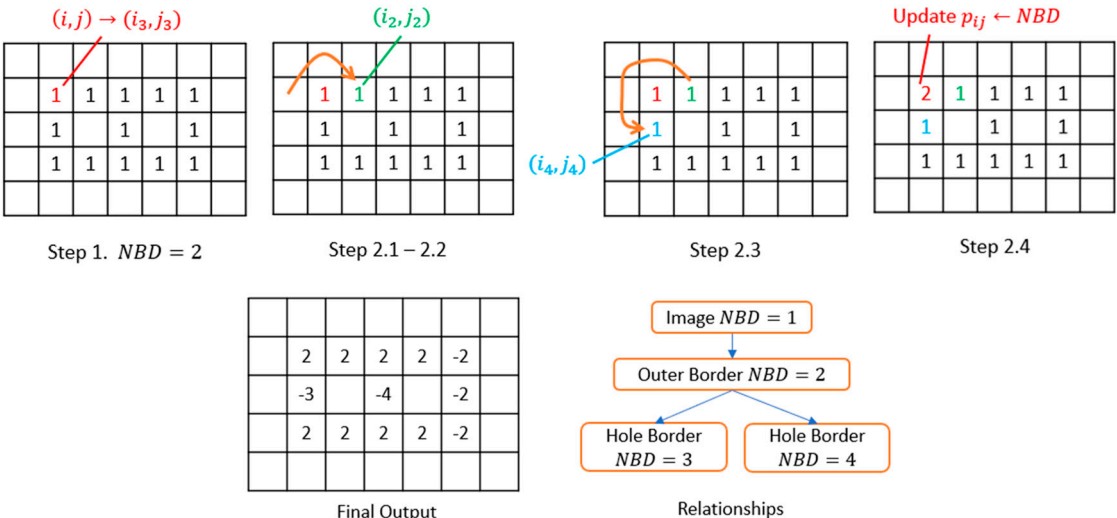

**Figure 6.** The introduction of Step 1, 2.1–2.4 and the introduction of the final output to the contour tracing algorithm.

Figure 6 show the contour tracing algorithm. By using this approach, the outer border or the contour of the arrow-like pavement marking can be found. However, due to uneven lighting or faded

markings, the detected contours are not closed curves, as shown in Figure 7b. The incomplete contours cannot help separate the pavement marking portion. To solve this problem, the dilation operation is performed before the contours are traced.

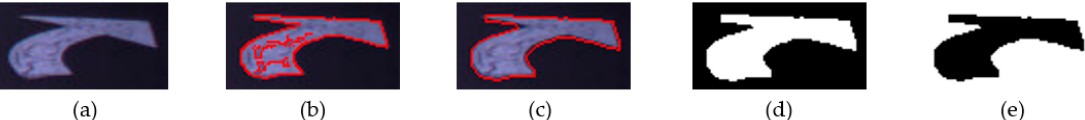

**Figure 7.** Results of the visibility analysis module. (**a**) Original patch, including the pavement marking; (**b**) Found contours without the dilation operation; (**c**) Found contours with the dilation operation; (**d**) Generated image mask for the marking; and (**e**) Generated image mask for the pavement.

Dilation is one of the morphological image processing methods, opposite to erosion [32]. The basic effect of the dilation operator on a binary image is the gradual enlargement of the boundaries of the foreground pixels so that the holes in the foreground regions would become smaller. The dilation operator takes two pieces of data as inputs. The first input is the image to be dilated, and the second input is a set of coordinate points known as a kernel. The kernel determines the precise effect of the dilation on the input image. It presumes that the kernel is a $3 \times 3$ square, with the origin at its center. To compute the dilation output of a binary image, each background pixel (i.e., 0-value) should be processed in turns. For each background pixel, if at least one coordinate point inside the kernel coincides with a foreground pixel (i.e., 1-value), the background pixel must be flipped to the foreground value. Otherwise, the next background pixel must be continually processed. Figure 8 shows the effect of a dilation using a $3 \times 3$ kernel. By using the dilation method before detecting the contours for the pavement marking patches, the holes in the markings are significantly eliminated, and the outer border becomes consistent and complete, which can be easily observed in Figure 7b,c.

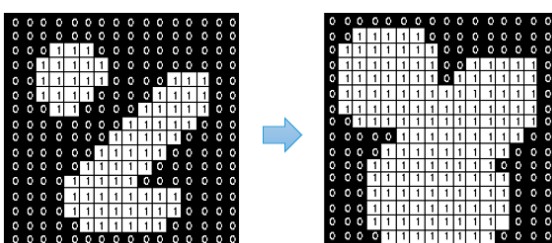

**Figure 8.** An example of the effect of the dilation operation (https://homepages.inf.ed.ac.uk/rbf/HIPR2/dilate.htm).

3.3.2. Construct Masks

Once the complete outer border of the pavement marking is obtained, the next step is to detach the pavement marking from the surrounding pavement. In practical scenarios, the pavement marking cannot be physically separated from the image patch due to its arbitrary shape. The most common way to achieve the target is to use masks to indicate the region segmentation. Since there are only two categories of objects, i.e., the pavement markings and the pavement, in this study, two masks had to be generated for each image patch.

Image masking is a non-destructive process of image editing that is universally employed in graphics software such as Photoshop to hide or reveal some portions of an image. Masking involves setting some of the pixel values in an image to 0 or another background value. Ordinary masks have only 1 and 0 values, and areas with a 0 value should be hidden (i.e., masked). Examples of masks generated for pavement markings are shown in Figure 7d,e.

### 3.3.3. Computing the Intensity Contrast

According to the pipeline of the visibility analysis module, the final step is to calculate the contrast between the pavement markings and the surrounding pavement. The straightforward way to determine the contrast value is to simply compute the difference between the average intensities of the markings and the pavement. However, this procedure does not adapt to the changes in the overall luminance. For instance, a luminance difference of 60 grayscales in a dark scenario (e.g., at night) should be more significant than the same luminance difference in a bright scenario (e.g., a sunny day). The human eyes sense brightness approximately logarithmically over a moderate range, which means the human visual system is more sensitive to intensity changes in dark circumstances than in bright environments [33]. Thus, in this study, the intensity contrast was computed using the Weber contrast method, the formula for which is:

$$Contrast(M,P) = \frac{\overline{I_M} - \overline{I_P}}{\overline{I_P}}, \ \overline{I_M} = \frac{\sum_{v \in Marking} I_v}{N_{Marking}}, \ \overline{I_P} = \frac{\sum_{v \in Pavement} I_v}{N_{Pavement}} \tag{3}$$

where $\overline{I_M}$ and $\overline{I_P}$ are the average intensity values of the pavement marking and the surrounding pavement, respectively, and the $N_{region}$ is the number of pixels in the specific region.

## 4. Experimental Validation of the Framework

### 4.1. Experiment Settings

Regarding the pavement marking detection model, it needs to be trained with a labelled dataset to enhance its performance. In this study, a Windows 10 personal computer with an Nvidia GeForce RTX 2060 Super GPU and a total memory of 16 GB was used to perform the training and validation procedures. The deep learning framework that was used to build, train, and evaluate the detection network is the TensorFlow platform, which is one of the most popular software libraries used for machine learning tasks [34].

On actual roads, left-turn markings are much more common than right-turn markings. This leads to an imbalanced ratio of the proportions of these two kinds of pavement markings in the training dataset. If a classification network is trained without fixing this problem, the model could be completely biased [35]. Thus, in this study, data augmentation was performed before the model was trained. Specifically, for each left-turn (right-turn) marking, the image was flipped along the horizontal axis to make the left-turn (right-turn) marking a new right-turn (left-turn) marking. By applying this strategy to the whole training dataset, the numbers of the two markings should be the same. An example of this data augmentation method is shown in Figure 9.

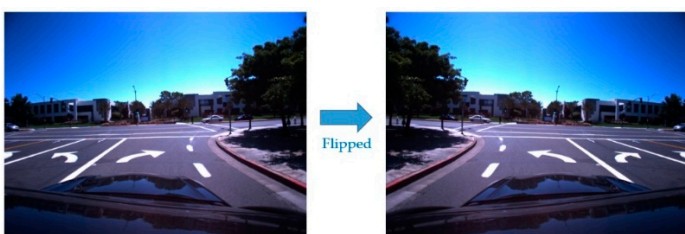

**Figure 9.** An example of data augmentation.

### 4.2. Model Training

The neural network is trained by first calculating the loss through a forward inference, and then updating related parameters based on the derivative of loss to make the predictions as accurate as possible. Therefore, the design of loss functions is significant. In the YOLOv3 algorithm, the loss

function has mainly three parts: the location offset of the predicted boxes, the deviation of the target confidence score, and the target classification error. The formula for the loss function is:

$$L(l, g, O, o, C, c) = \lambda_1 L_{loc}(l, g) + \lambda_2 L_{conf}(o, c) + \lambda_3 L_{cla}(O, C), \tag{4}$$

where $\lambda_1 \sim \lambda_3$ refers to the scaling factors.

The location loss function uses the sum of the square errors between the true offset and the predicted offset, which is formulated as:

$$L_{loc}(l, g) = \sum_{m \in \{x, y, w, h\}} (\hat{l}^m - \hat{g}^m)^2 \tag{5}$$

where $\hat{l}$ and $\hat{g}$ represent the coordinate offsets of the predicted bounding box and the referenced bounding box, respectively. Both $\hat{l}$ and $\hat{g}$ have four parameters: $x$ for the offset along the $x$-axis, $y$ for the offset along the $y$-axis, $w$ for the box width, and $h$ for the box height.

The target confidence score indicates the probability that the predicted box contains the target, which is computed as:

$$L_{conf}(o, c) = -\sum (o_i ln(\hat{c}_i) + (1 - o_i)ln(1 - \hat{c}_i)). \tag{6}$$

The function $L_{conf}$ uses the binary cross-entropy loss, where $o_i \in \{0, 1\}$ indicates whether the target actually exists in the predicted rectangle $i$. The 1 value means yes, and the 0 value means no. $c_i \in [0, 1]$ denotes the estimated probability that there is a target in the rectangle $i$.

The formulation of the target classification error in this study slightly differs from that in the YOLOv3 network. In the YOLOv3 network, the authors still used the binary cross-entropy loss function, as the author thought the object was possibly classified into more than one category in complicated reality scenes. However, in this study, the categories of the pavement markings were mutually exclusive. Thus, the multi-class cross-entropy loss function was used to measure the target classification error, the mathematical expression of which is:

$$L_{cla}(O, C) = -\sum_{i \in pos} \sum_{j \in cla} (O_{ij} ln(\hat{C}_{ij}) + (1 - O_{ij})ln(1 - \hat{C}_{ij})), \tag{7}$$

where $O_{ij} \in \{0, 1\}$ indicates if the predicted box $i$ contains the object $j$, and $\hat{C}_{ij} \in [0, 1]$ represents the estimated probability occurring in the aforementioned event.

Pan and Yang (2010) found that in the machine learning field, the knowledge gained while solving one problem can be applied to another different but related problem, which is called transfer learning [36]. For instance, the knowledge obtained while learning to recognize cars could be useful for recognizing trucks. In this study, the pavement marking detection network was not trained from scratch. Instead, a pre-trained model learning to recognize objects in the MS COCO dataset was used for the initialization. The MS COCO dataset, published by Lin et al., contains large-scale object detection data and annotations [37]. The model pre-trained from the COCO dataset can provide the machine with some general knowledge on object detection tasks. Starting from the pre-trained network, a fine-tuned procedure is conducted by feeding the collected data to the machine to make it capable of recognizing pavement markings. The total training process runs for a total of 50 epochs.

With the help of the TensorBoard integrated into the TensorFlow platform, users can monitor the training progress in real time. It can export figures to indicate the trends of specific parameters or predefined metrics. Figure 10 shows the trend of three different losses during the training process. The figure shows a decreasing trend for all the losses.

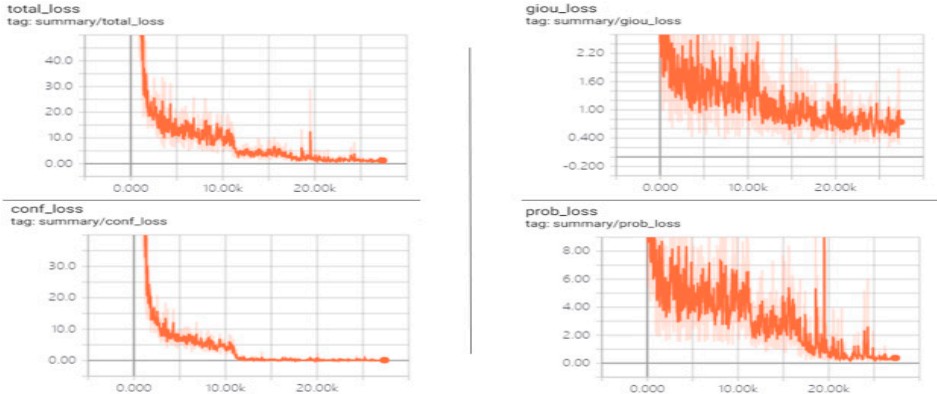

**Figure 10.** The trends of various loss functions during the training process monitored by TensorBoard.

*4.3. Model Inference and Performance*

After the training, the produced model is evaluated on the testing dataset. At the end of each training epoch, the network structure and the corresponding parameters are stored as the checkpoint file. For the evaluation, the checkpoint file with the least loss is chosen to be restored. The testing sample images are directly fed to the model as the inputs, and then the machine will automatically detect and locate the pavement markings in the image. Once the arrow-like pavement markings are recognized in the image, the detected areas are extracted to perform the visibility analysis. In this study, the function of the visibility analysis module was integrated into the evaluation of the pavement marking detection module. Thus, for each input image, the model drew the predicted bounding boxes, and added text to indicate the estimated category, the confidence score, and the contrast score on the image. Some examples of the evaluation of testing images are shown in Figure 11.

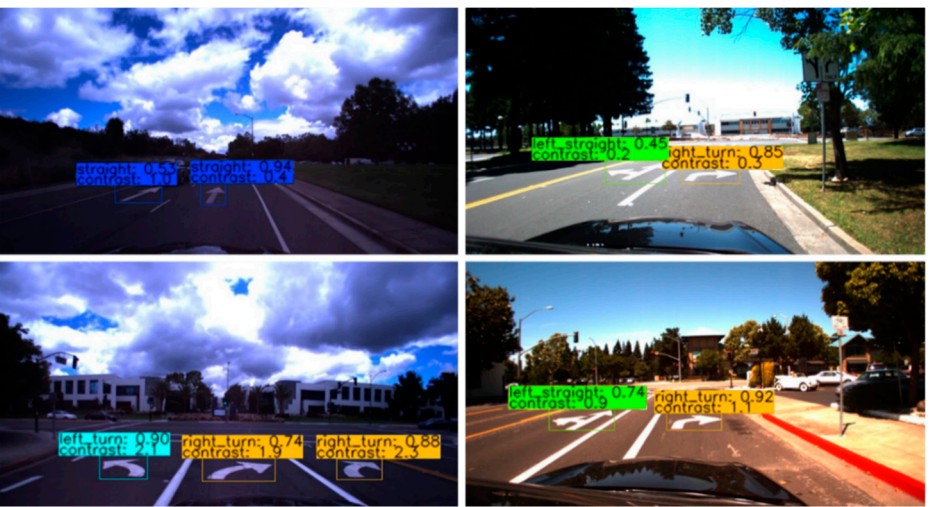

**Figure 11.** Visual results were evaluated on the testing samples.

From the figure, it can be seen that most of the pavement markings are correctly located and classified, and the contrast value provides a good measure of the visibility of the markings. The two subfigures on the left belong to the cloudy scenario, and the two on the right represent the sunny case. The pavements in the two subfigures on the left are both dark; but due to the poor marking condition, the contrast values of the top subfigure (i.e., 1.0, 0.4) are much lower than those at the bottom (i.e., 2.1, 1.9, 2.3). It can be observed that the pavement markings in the bottom subfigure are much more recognizable than those in the top subfigure, which validates the effectiveness of the contrast value for analyzing the visibility of pavement markings. Similarly, for the two subfigures on

the right, all the detected pavement markings are in good condition; nevertheless, the contrast value of the bottom subfigure (i.e., 0.9, 1.1) is higher than that of the top subfigure (i.e., 0.2, 0.3), because the pavement in the bottom image is darker. This means the markings in the bottom-right subfigure are easier to identify than those in the top-right subfigure. The high brightness of the pavement could reduce the visibility of the markings on it, as the markings are generally painted white.

For the quantitative evaluation of the performance of the pavement marking detection model in this study, the mean average precision (mAP) was used. The results of the object detection system were divided into the following four categories by comparing the estimation with the reference label: true positives (TP), true negatives (TN), false positives (FP), and false negatives (FN). Table 1 defines these four metrics.

**Table 1.** Four categories of the metrics.

|  | **Positive Predication** | **Negative Prediction** |
|---|---|---|
| Positive Label | TP | FN |
| Negative Label | FP | TN |

The precision refers to the proportion of the correct results in the identified positive samples, and the recall denotes the ratio of the correctly identified positive samples to all the positive samples. The formulas for these two metrics are as follows.

$$Precision = \frac{TP}{TP + FP} \qquad\qquad Recall = \frac{TP}{TP + FN} \tag{8}$$

To determine if a prediction box correctly located the target, an IoU threshold was predefined before the model was evaluated. For as long as the IoU value between the estimated bounding box and the ground truth was bigger than the threshold, this prediction was considered a correct detection. When the threshold value was adjusted, both the precision and the recall changed. As the threshold decreased, the recall value continued to increase, and the accuracy decreased after reaching a certain level. According to this pattern, the precision-recall curve, i.e., the PR curve, was drawn [38]. The AP value refers to the area under the PR curve, and the mAP value indicates the average AP among the multiple categories.

Figure 12 shows the results of the quantitative validation of the detection model on the testing dataset. As shown in the top-left subfigure, there are 203 sample images and 223 pavement marking objects included in the evaluation dataset. It can be seen that the distribution of different pavement markings is imbalanced. Thus, collecting more images and enlarging the dataset are the future study orientations for this proposal. The bottom-left subfigure demonstrates the number of true/false predictions upon the testing samples for each category, where the red portion represents the false predictions and the green potion refers to the true predictions. Given the number shown in the figure, it can be surmised that the detection model is working properly since most of the identified pavement markings were correctly classified. The right subfigure provides the average precision values for each category. The mAP value can reflect the overall performance of the detection module. However, the low mAP value indicates that there are some spaces to further improve the model.

From the validation results on testing samples, it is observed that some left-turn and right-turn markings are misclassified as the other category. By exploring the whole project, the reason causing this issue is finally found: a code issue. Since YOLOv3 is a representative framework in the object detection field, there are many open-source implementation codes. In this project, the detection model upon pavement markings is also trained with the open-source codes. Within the data preprocessing step of the codes, the author randomly chooses some training samples and flips them horizontally to enhance the diversity of the training data. Actually, this is a common and useful operation to achieve data augumentation. However, it does not fit for this pavement marking detection task. For general objects, the horizontal flipping would not change its category so that this operation is valid. But in terms of

pavement markings, the flip process may transform the marking into another type, e.g., left-turn to right-turn. Thus, the flip operation within the codes generates wrong training samples, misleads the machine and hinders the performance of the model.

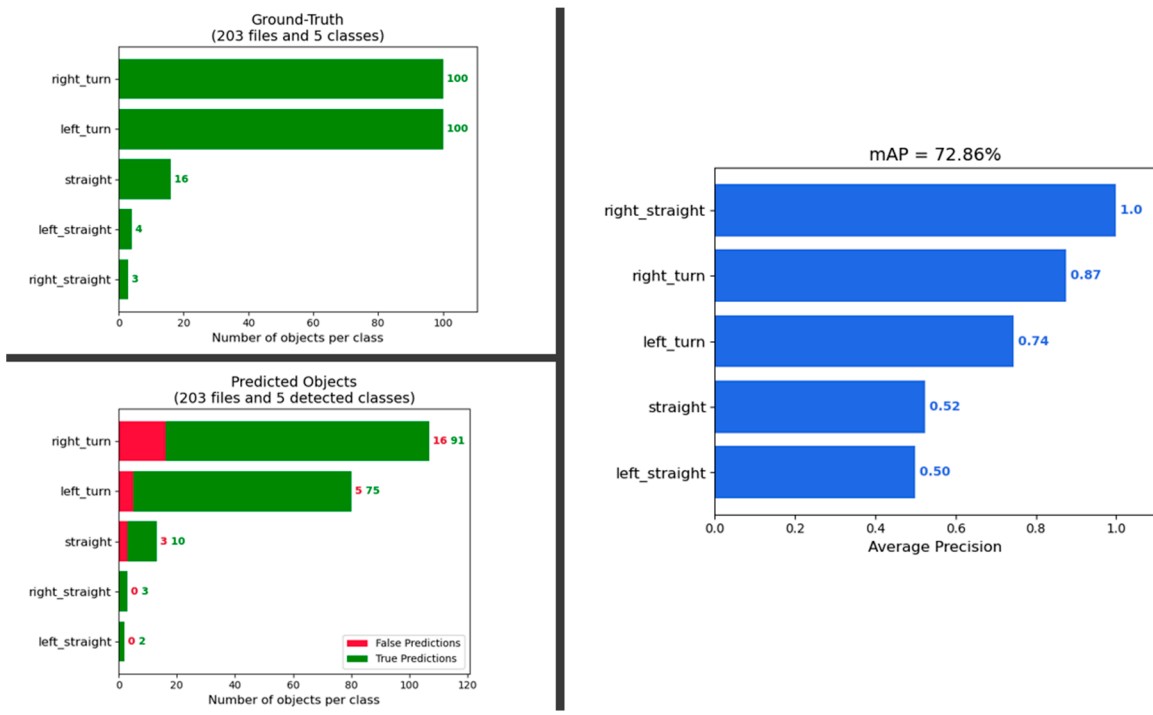

**Figure 12.** The quantitative evaluation information on the testing dataset of the trained model.

By removing the codes and re-training the model, the new quantitative validation results are shown in Figure 13. Comparing the Figures 12 and 13, the performance of the model is greatly enhanced, i.e., there is a 24% increment on the mAP value. The evaluation results fully prove the effectiveness of the YOLOv3 model in the pavement marking recognition task.

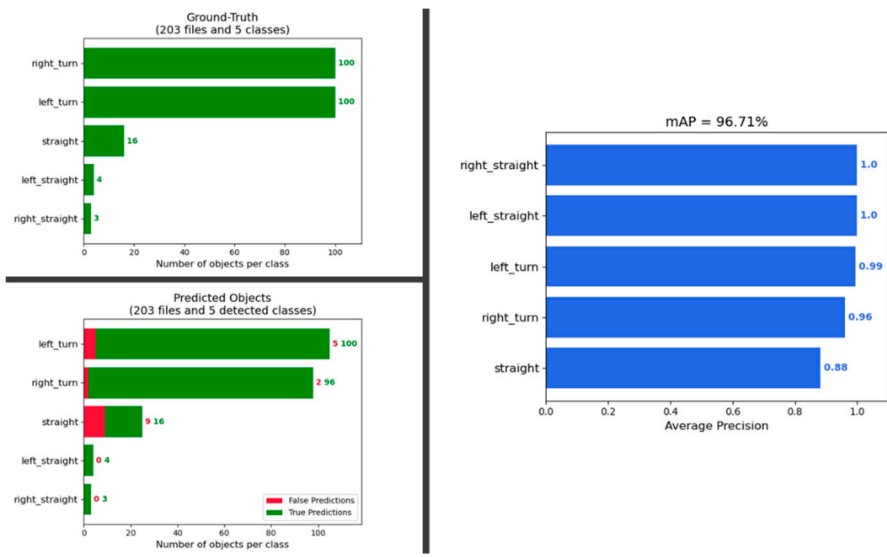

**Figure 13.** The quantitative evaluation information on the testing dataset of the improved model.

## 5. Conclusions

To identify issues with the detection and visibility of pavement markings, relevant studies were reviewed. The automated condition analysis framework for pavement markings using machine learning technology was proposed. The framework has three modules: a data processing module, a pavement marking detection module, and a visibility analysis module. The framework was validated through a case study of pavement marking training data sets in the U.S. From the quantitative results in the experimental section, the precision of the pavement marking detection module was pretty high, which fully validates the effectiveness of the YOLOv3 framework. Meanwhile, observing the visual results, all the pavement markings are correctly detected with the rectangle boxes and classified with the attached text in the road-scene images. In addition, the visibility metric of pavement markings was defined and the visibility within the proposed framework was confirmed as an important factor of driver safety and maintenance. The computed visibility values were also attached besides the detected pavement markings in the images. If the proposed study is used properly, pavement markings can be detected accurately, and their visibility can be analyzed to quickly identify places with safety concerns.

From the distribution of the testing samples, it can be inferred that the proportions of the straight markings, the right straight markings, and the left straight markings could be very low. Enlarging and enriching the training dataset could be a goal for future research.

**Author Contributions:** Conceptualization, K.K., T.K. and J.K.; Data curation, K.K.; Formal analysis, K.K., D.C. and C.P.; Investigation, D.K. and J.K.; Methodology, K.K., D.K., T.K. and J.K.; Project administration, T.K.; Resources, T.K.; Software, K.K., D.C., C.P. and T.K.; Writing—review & editing, K.K. and T.K. All authors have read and agreed to the published version of the manuscript.

**Funding:** This research was supported by a grant (KICT 2020-0559) from the Remote Scan and Vision Platform Elementary Technology Development for Facility and Infrastructure Management funded by KICT (Korea Institute of Civil Engineering and Building Technology) and a grant (20AUDP-B127891-04) from the Architecture and Urban Development Research Program funded by the Ministry of Land, Infrastructure, and Transport of the Korean government.

**Conflicts of Interest:** The authors declare no conflict of interest.

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
