# Peer review of "Development of an Automated Visibility Analysis Framework for Pavement Markings Based on the Deep Learning Approach"

_remotesensing, doi:10.3390/rs12223837_

Round 1
Reviewer 1 Report
Dear Authors:
congratulations, your work presents a proposal for one of the important tasks in the driver assistant systems or for self-driven cars, it's promising. However, most part of the manuscript contains report in a didactical view and it turns hard to follow. I understand the paper's contribution, but this in the introduction is not clear, a more precise description of the contribution is needed. With respect to related studies, I enjoyed read it, however, in some parts of the methodology section you are still presenting related works, which has to be tackled in the previous section. In 3.1.2 there is unnecessary information about the characteristics of VoTT. The steps in 3.3.1 need improvements, most of the details there made me lose the path, I suggest trying with the algorithm for those steps.
Concerning the results, I am still not clear how many images were used for testing (pavement marking types). I saw that your proposal was evaluated on different environmental conditions, even so, they are not enough to validate the system, at least I suggest the usage of Gaussian noise and then check the results. Fig. 12 is unclear, try using another plot or just tables.
Again, your contribution uses a solid Deep Learning model, and I realized that the scientific process is properly organized, but the presentation in the manuscript makes, in most of the parts, difficult to follow.
Author Response
Response to Reviewer 1 Comments
congratulations, your work presents a proposal for one of the important tasks in the driver assistant systems or for self-driven cars, it's promising. However, most part of the manuscript contains report in a didactical view and it turns hard to follow.
Point 1: I understand the paper's contribution, but this in the introduction is not clear, a more precise description of the contribution is needed.
Response 1:
Thank you for your feedback. The authors have revised and added the following sentences in the introduction (line 46 – 50):
[The data processing module includes data acquisition and data annotation, which provides a clear and accurate dataset for the detection module to train. In the pavement marking detection module, a framework named YOLOv3 is used for training to detect and localize pavement markings. For the visibility analysis module, the contour of each pavement marking is clearly marked, and each contrast intensity value is also provided to measure visibility.]
Point 2: With respect to related studies, I enjoyed read it, however, in some parts of the methodology section you are still presenting related works, which has to be tackled in the previous section.
Response 2:
Thank you for your feedback. This study consists of new methods with several previous works as key steps of the research. The authors believe it would be more appropriate to explain those previous works with the methods that our study propose in the methodology section (particularly line 168-188 of the original manuscript) to keep consistent flows and structure of our logic. The authors revised the section 3.1.2, and removed unnecessary sentences. Please see the revised paragraph below (line 158-161 of the revised manuscript).
[VoTT is a powerful open-source labeling software released by Microsoft [19]. This software provides a technique for automatic labeling based on the pre-trained network, which can significantly reduce the workload for annotations. It also supports many formats of the exported annotation results, which make the labeled sample set suitable for various deep learning development frameworks. Figure 2 shows an example of the labeling process. ]
Point 3: In 3.1.2 there is unnecessary information about the characteristics of VoTT.
Response 3:
Thank you for your feedback. The authors have revised and deleted several sentences. Please see the revised section below (line 158-161 of the revised manuscript):
[This software provides a technique for automatic labeling based on the pre-trained network, which can significantly reduce the workload for annotations. It also supports many formats of the exported annotation results, which make the labeled sample set suitable for various deep learning development frameworks.]
Point 4: The steps in 3.3.1 need improvements, most of the details there made me lose the path, I suggest trying with the algorithm for those steps.
Response 4:
Thank you for your comment. The authors have revised the section 3.3.1 and added more detailed descriptions about variables and reorganized the structure of contour tracing algorithm. Please see the revised paragraph below (line 297-333 of the revised manuscript)
[Two variables, NBD(newest border number),LNBD (last newest border number), are created to record the relationship between the pixels during the scanning process. The algorithm uses the row-by-row and left-to-right scanning schemes to process each NBD and LNBD, where p_ij>0.]
Point 5: Concerning the results, I am still not clear how many images were used for testing (pavement marking types). I saw that your proposal was evaluated on different environmental conditions, even so, they are not enough to validate the system, at least I suggest the usage of Gaussian noise and then check the results.
Response 5:
Thank you for your comment. The authors have used more than 1,000 images for training and 200 images for testing, and the authors believe that training-testing ratio is sufficient for evaluating. Several sentences have been added in the revised manuscript, please see the revised paragraph below (line 150-152 of the revised manuscript).
[After screening all the video data, more than 1,000 high-quality pictures were intercepted, of which about 200 were used for testing, and the remaining pictures were used for training, which maintaining a good training-testing ratio.]
The authors didn’t classify environmental condition for each image, and the authors believe adding noise may influence visibility analysis module that we study proposes, the authors will definitely consider adding Gaussian noise for the future study.
Point 6: Fig. 12 is unclear, try using another plot or just tables.
Response 6:
Thank you for your feedback. The authors have solved the resolution problem and improved the quality of Fig. 12 (line 498 of the revised manuscript). In addition to Fig. 12, the authors have added Fig. 13 for better explanation.
Again, your contribution uses a solid Deep Learning model, and I realized that the scientific process is properly organized, but the presentation in the manuscript makes, in most of the parts, difficult to follow.
Thanks for your comments.
Sincerely

Reviewer 2 Report
Dear authors
This paper sets out to describe a methodology for Development of an Automated Visibility Analysis Framework for Pavement Markings based on the Deep Learning Approach
Overall, the article is technically correct. However, there is room for improvement.
Revision:
- The phrases described in lines [271 to 273] and [367 to 369], lack an appropriate reference, respectively.
- It is necessary to improve the presentation quality of the graphs in figures 10 and 12.
- On page 15, it is suggested to elaborate more on the information provided in Figure 12; this could also be employed to extend the relevance of the proposal.
- A weak point in this paper is the conclusion; it must be improved. It is suggested to expand the comments in the results.
- It would be important to compare the results obtained with other proposals in the academic literature.
Author Response
Response to Reviewer 2 Comments
Dear authors
This paper sets out to describe a methodology for Development of an Automated Visibility Analysis Framework for Pavement Markings based on the Deep Learning Approach
Overall, the article is technically correct. However, there is room for improvement.
Revision:
Point 1: The phrases described in lines [271 to 273] and [367 to 369], lack an appropriate reference, respectively.
Response 1:
Thank you for your feedback. The authors have added appropriate reference to support the statements. Please see the added reference below:
[
[31] Klein, Stanley A., Thom Carney, Lauren Barghout-Stein, and Christopher W. Tyler. "Seven models of masking." In Human Vision and Electronic Imaging II, vol. 3016, pp. 13-24. International Society for Optics and Photonics, 1997
]
Point 2: It is necessary to improve the presentation quality of the graphs in figures 10 and 12.
Response 2:
Thank you for your feedback. The authors have replaced the original figures with the better resolution. Please see the replaced figures for Fig. 10 and Fig. 12 (line 446 and 498 of the revised manuscript).
Point 3: On page 15, it is suggested to elaborate more on the information provided in Figure 12; this could also be employed to extend the relevance of the proposal.
Response 3:
Thank you for your valuable suggestion. The authors have added some sentences to illustrate the detailed information of quantitative results that showed in Fig. 12. The revised paragraphs for this figure are shown as follows (line 501-510 of the revised manuscript).
[As shown in the top-left subfigure, there are 203 sample images and 223 pavement marking objects included in the evaluation dataset. The distribution of different pavement markings is imbalanced. Thus, collecting more images and enlarging the dataset are the future study orientations for this proposal. The bottom-left subfigure demonstrates the number of true/false predictions upon the testing samples for each category, where the red potion represents the false predictions and the green potion refers to the true predictions. Given the number shown in the figure, it can be surmised that the detection model is working properly since most of the identified pavement markings were correctly classified. The right subfigure provides the average precision values for each category. The mAP value can reflect the overall performance of the detection module. However, the low mAP value indicates that there isare some spaces to further improve the model.]
Point 4: A weak point in this paper is the conclusion; it must be improved. It is suggested to expand the comments in the results.
Response 4:
Thank you for your suggestion. The authors have updated the contents in the conclusion section to provide better illustration on the effectiveness of the proposed framework. The revised paragraphs in the section are shown as below (line 534-541 of the revised manuscript).
[From the quantitative results in the experimental section, the precision of the pavement marking detection module was pretty high, which fully validate the effectiveness of the YOLOv3 framework. Meanwhile, observing the visual results, all the pavement markings are correctly detected with the rectangle boxes and classified with the attached text in the road-scene images. In addition, the visibility metric of pavement markings was defined and visibility within the proposed framework was confirmed as an important factor of driver safety and maintenance. The computed visibility values were also attached besides the detected pavement markings in the images.]
Point 5: It would be important to compare the results obtained with other proposals in the academic literature.
Response 5:
Thank you for your comment. The authors have fixed some issues of the previous model and obtained a pretty better model (mAP reaches 96.7%). The evaluation results of the new model are shown in Fig. 13 (line 523 of the revised manuscript). The performance of the new model can show the effectiveness of the pavement marking detection module. Unfortunately, none of the previous papers, the authors have reviewed, performed condition analysis that could be directly compared with this study’s findings.
Thanks for your comments.
Sincerely

Reviewer 3 Report
In this paper, an automated condition analysis framework for pavement markings using machine learning technology was constructed, consisting of three modules: a data processing module, a pavement marking detection module, and a visibility analysis module. The related technologies are introduced detailedly, but the experiments were thin (just arrow-like pavement markings). Main considerations about the study are shown:
- The spelling and grammar mistakes should be revised, such as “the most studied” (line 169) and so on.
- The abbreviation of specialized vocabulary first appeared in the paper should list the full name, such as NBD and LNBD in line 290 and line 291.
- All the variables in the formula should be explained what they are standing for.
- The dataset should be introduced detailedly, such as the volume and samples.
- There should be experimental comparisons to proof the superiority of your strategy.
Author Response
Response to Reviewer 3 Comments
In this paper, an automated condition analysis framework for pavement markings using machine learning technology was constructed, consisting of three modules: a data processing module, a pavement marking detection module, and a visibility analysis module. The related technologies are introduced detailedly, but the experiments were thin (just arrow-like pavement markings). Main considerations about the study are shown:
Point 1: The spelling and grammar mistakes should be revised, such as “the most studied” (line 169) and so on.
Response 1:
Thank you for letting us know. The authors have revised grammar mistakes of the manuscript.
Point 2: The abbreviation of specialized vocabulary first appeared in the paper should list the full name, such as NBD and LNBD in line 290 and line 291.
Response 2:
Thank you for your feedback. The authors have provided the full forms of abbreviations that were used in the manuscript (line 298)
[Two variables, Newest Border Number (NBD), Last Newest Border Number (LNBD), are created to record the relationship between the pixels during the scanning process.]
Point 3: All the variables in the formula should be explained what they are standing for.
Response 3:
Thank you for your feedback. The authors have checked all the formulas and added the description for the equations (line 215-217 of the revised manuscript).
[Suppose the possibility of the existance of the object is Pr(Obj), and the IoU value between the predicted box and the reference box is IoU(pred,truth), the formula for the confidence score is shown as Eq. 1.]
Point 4: The dataset should be introduced detailedly, such as the volume and samples.
Response 4:
Thank you for your feedback. The authors have added detailed information about the dataset in the section 3.1.1. Data Acquisition.
[After screening all the video data, more than 1,000 high-quality pictures were intercepted, of which about 200 were used for testing, and the remaining pictures were used for training, which maintaining a good training-testing ratio.]
Point 5: There should be experimental comparisons to proof the superiority of your strategy.
Response 5:
Thank you for your comment. The authors have fixed issues of the previous model and obtained a much better accuracy of the model (mAP reaches 96.7%). The evaluation results of the new model are shown in Fig. 13 (line 524 of the revised manuscript). The performance of the new model can show the effectiveness of the pavement marking detection module. Unfortunately, none of the previous papers, the authors have reviewed, performed condition analysis that could be directly compared with this study’s findings.
Thanks for your comments.
Sincerely

Reviewer 4 Report
The paper addresses the task which is important for many countries that have well developed infrastructure and large territories. Automation of road control is important and allows getting essential benefits. The paper introduces a quite novel approach to markinf detection based on modern machine learning technologies. The paper is well balanced and well written.
Concerning possible improvements, I'd like to mention the following:
1) Has some essential dependence of detection characteristics on weather conditions been observed? If yes, are there some practical recommendations?
2) How general is the proposed algorithm? Can it be used for marking detection in other countries of neural network retraining should be used? Will it be necessary to change other parts of the algorithm?
3) Do the contrast values in Fig. 11 relate to Equation 3?
4) Does a used camera influence the results of detection? If yes, how large is this influence? Are there recommendations to used cameras?
Author Response
Response to Reviewer 4 Comments
The paper addresses the task which is important for many countries that have well developed infrastructure and large territories. Automation of road control is important and allows getting essential benefits. The paper introduces a quite novel approach to markinf detection based on modern machine learning technologies. The paper is well balanced and well written.
Concerning possible improvements, I'd like to mention the following:
Point 1: Has some essential dependence of detection characteristics on weather conditions been observed? If yes, are there some practical recommendations?
Response 1:
Thank you for your questions. Unfortunately, the scope of this study has not covered detection characteristics on weather condition. However, it is excellent idea for the follow-up study. Appreciate it.
Point 2: How general is the proposed algorithm? Can it be used for marking detection in other countries of neural network retraining should be used? Will it be necessary to change other parts of the algorithm?
Response 2:
Thank you for your question. The answer would be “Yes” and/or “No” depending on pavement marking styles of other countries. Although it is different style, with adding appropriate training dataset, the proposed algorithm can be used without any further modification. If countries use same pavement marking styles, the proposed algorithm can be applied right away.
Point 3: Do the contrast values in Fig. 11 relate to Equation 3?
Response 3:
Thank you for your question. Yes, the contrast values in Fig. 11 were calculated using Eq. 3.
Point 4: Does a used camera influence the results of detection? If yes, how large is this influence? Are there recommendations to used cameras?
Response 4:
Thank you for your question. The camera, in terms of resolutions, does not affect much the results of detecting pavement marking using the proposed algorithm. However, if capture images are not in focus (e.g., blurred images), it may affect the detection ratio of pavement markings. It will be very interesting to see how image qualities will affect (or influence) results of detection. The authors will conduct the follow-up study on that.
Thanks for your comments.
Sincerely

Round 2
Reviewer 3 Report
I think the paper is suitable for publication now.